# Strategy for Ensuring the Metrological Traceability of Nanoparticle Size Measurements by SEM

**DOI:** 10.3390/nano14110931

**Published:** 2024-05-25

**Authors:** Nicolas Feltin, Alexandra Delvallée, Loïc Crouzier

**Affiliations:** Laboratoire National de Métrologie et d’Essais (LNE), 29 Avenue Roger Hennequin, CEDEX, 78197 Trappes, France; alexandra.delvallee@lne.fr (A.D.); loic.crouzier@lne.fr (L.C.)

**Keywords:** SEM calibration, traceable nanoparticle size measurements

## Abstract

The concept of measurement traceability is crucial for ensuring the data reliability and the comparability of measurement results provided by different instruments and operators. In the field of nanoparticle metrology, determining the size of nanoparticles using electron microscopy-based techniques remains a real challenge. In laboratory settings, the establishment of traceability regarding the instrument calibration procedures, the assessment of uncertainties associated with instruments/operators/samples/environments, as well as the complexities related to electron–sample interactions, are often neglected. In this article, we describe the calibration procedure set up at the LNE (Laboratoire National de métrologie et d’Essais) and propose an evaluation method for determining the uncertainties in measuring nanoparticle size by SEM (Scanning Electron Microscopy). This study investigates the impact of the energy of the primary electrons (PEs) generated by the electron beam and accelerating voltage on the reliability of size measurements. The convolution between the signals coming from a nanoparticle and the substrate on which the particle is deposited induces edge effects that can have a negative impact on the measurement results. Finally, a diagram describing the various stages involved in establishing traceability for SEM measurements of nanoparticle size is proposed to facilitate the work of future operators.

## 1. Introduction

According to the International Vocabulary of Metrology (VIM), metrological traceability refers to “the property of a measurement result whereby the result can be related to a reference through a documented unbroken chain of calibrations, each contributing to the measurement uncertainty” [1]. In other words, traceability establishes a direct link between the measurement result and the SI (Système International) unit to which it is attached. The international system of units is the only common reference for all laboratories and industries around the world. The calibration of measurement equipment and the interpretation of the associated data are fundamental steps in ensuring the reliability of scientific results. These steps makes characterization results comparable between laboratories throughout the world.

Nanometrology, the science of measurement at the nanoscale, has a crucial role in the manufacturing of nanomaterials and nanodevices with a high degree of accuracy and reliability [2,3,4]. A nano-object behaves differently from corresponding bulk material. The associated metrology requires the development of specific protocols to properly characterize and understand these nanomaterials. Generally, industrial and academic institutions focus on length metrology because the definition of a nanomaterial is primarily based on the notion of size. However, nanoparticles are three-dimensional objects and the development of dimensional nanometrology is necessary for reinforcing quality control in the manufacturing process at the nanoscale. It can also provide reliable information related to the potential health risks of nanoparticles. In addition, from an R&D perspective, length metrology at the nanoscale makes it possible to establish a link between NP size and expected performance.

In this context, microscopy-based techniques, such Transmission Electron Microscopy (TEM) or secondary electron imaging in Scanning Electron Microscopy (SEM), are considered reference techniques for measuring nanoparticle size. The physics of SEM is complex but this surface technique is more versatile than TEM and does not require an electron transparent sample. In SEM, a highly focused electron beam (e-beam) scans the sample, generating backscattered or secondary electrons (SEs) which are collected by detectors that measure their intensity. Modern Scanning Electron Microscopes have a resolution of roughly 1 nm depending on the spot diameter of the e-beam. SEM-based imaging techniques are essentially counting methods and are particularly suitable for determining the size distribution of an NP population. The image acquisition process must be preceded by a specific sample preparation whose objective is to effectively disperse the particles on the substrate in order to limit the agglomerate formation inducing measurement errors [5]. Then, the isolated particles are measured one by one from a SEM image and the number of particles to be counted to construct a size distribution histogram statistically representative of the population depends on the targeted uncertainty, the polydispersity and the nature of the size distribution (Gaussian, lognormal…) [6]. The results, median/mode/mean size, are extracted from this histogram. The construction of the latter is essential to detect NP multimodal populations. In this case, each mode is fitted by a Gaussian/lognormal distribution for determining the characteristic parameters [5]. The most frequently used measurand in SEM measurements is the equivalent circular diameter (ECD), *D*_eq_, representing the diameter of a circle that occupies the same two-dimensional surface area as the object to be measured. It is similar to the real physical diameter in the case of a perfect sphere. But, when the shape of the particle differs from the sphere, *D*_eq_ can be replaced by *D*_MinFeret_ and *D*_MaxFeret_ corresponding, respectively, to the shortest and the longest dimensions of the particle whatever its orientation is. *D*_MinFeret_ is particularly well suited to characterise the *nano* nature of a substance, as the definition refers to the smallest dimension of the object being analysed.

Like any metrologist, an operator performing NP size measurements wants precise and accurate results, but precision and accuracy are two separate and distinct concepts. Precision is often defined as repeatability or the spread of values calculated after repeating the measurements several times on the same sample. Precision is globally directly linked to the following four factors: (i) instrument, (ii) operator, (iii) sample and (iv) environment. To investigate the influence of these factors, the metrological performance of the microscope must be characterised; several operators must carry out the same measurements on the same sample to assess the operator effect and special attention should be paid to sample preparation. In the case of electron microscopy-based techniques (EM), the sample is introduced into a vacuum chamber with a vacuum level of at least (5 × 10^−6^ mbar), and thus the environment effect can be considered as insignificant.

Moreover, the repeatability of the measurements is dependent on the operator’s ability to optimise the setup parameters for the appropriate nanoscale magnification range by adjusting the working distance, e-beam current and spot diameter. As explained in the following sections, accelerating voltage (AV) is also a key parameter for accuracy and should be studied with care.

The first step in NP analysis is calibration. A reference measurement procedure must be followed to determine the trueness of measurement values. Trueness and precision guarantee the accuracy of the measurement. A Scanning Electron Microscope calibration involves 2D calibration gratings, containing lateral periodic patterns with a defined mean pitch or certified reference nanoparticles [3,7]. Many calibration gratings and a few reference nanoparticles (latex, silica and gold) are available on the market with a certified pitch/mean diameter for electron microscopy. The characteristics of reference materials and certified reference materials are described in ISO Guides 30–33 [8,9,10]. The traceability to the SI unit of the gratings is ensured by direct measurement techniques such as mAFM (metrological Atomic Force Microscope) [11,12] or by indirect techniques using deep UV laser diffraction which gives the reference values for the mean pitch along the X and Y directions with a few nanometres of uncertainty [13,14]. This traceability chain using commercially available transfer standards and reference techniques (mAFM or optical diffractometer) establishes metrological traceability at the nanoscale to SI units. Linking the measurements performed by an operator and the unit definition of the length ensures the reliability of NP size results and ensures that these results are comparable to measurements made in other laboratories, institutes or companies.

In this paper, we show that accurate measurements of nanoparticles cannot be achieved by simply calibrating a microscope using the techniques described above, but the calibration process must also be accompanied by a model simulating the electron–matter interaction. In fact, a strong dependence between acceleration voltage and signal shape as a function of both particle size and chemical composition has been observed in previous studies using SE detectors [15] in SEM and transmitted electron detectors [16] in TSEM mode.

In the first section of this paper, we detail the SEM calibration procedure followed by the French Metrology Institute, the LNE (Laboratoire National de métrologie et d’Essais), using an mAFM and transfer standards as well as the method to determine the uncertainty value linked to the NP size measurements.

Then, we study the influence of a key parameter, accelerating voltage, on size measurements. We show that the shape of the measured SEM profile depends on the AV values, NP size, NP chemical composition and the nature of the substrate on which the particle is deposited. We demonstrate that without modelling of the SEM signal, measurement errors can occur.

Finally, we present a diagram summarising the various steps involved in obtaining reliable, traceable and therefore comparable measurements of nanoparticle size.

## 2. Materials and Methods

### 2.1. Material Preparation

In order to highlight the differences in NP behaviour linked to their size and their chemical composition during electron–matter interactions induced by the SEM e-beam, four samples of metal, oxide or polymer nanoparticles were studied. The first sample, FD-101b is a certified reference material (CRM) provided by the Joint Research Centre (JRC, Brussels, Belgium) and composed of a bi-modal population of silica NPs well separated in size (nominally 40 nm for the first population and nominally 80 nm for the second population). The FD-101b reference material report records a number-weighted modal equivalent circular diameter (ECD), *D_EC__*_modal_, of (83.7 ± 2.2) nm, as measured by EM (the associated expanded uncertainty corresponds to a 95% coverage probability, so to a coverage factor *k* = 2). The second NP population with a mean dimension close to 40 nm is not certified. The sample FD-304 is also a CRM provided by the JRC but the reference value was only determined by Dynamic Light Scattering (DLS). The certificate reports an indicative value for electron microscopy techniques of 27.8 nm with a 1.5 nm associated uncertainty (*k* = 2) [17]. The value recorded in the certificate is a modal diameter. Therefore, we have chosen to focus on the mode rather than the mean value throughout the article and all measurements are reported as modes. The third sample is composed of a polystyrene latex (PSL) bimodal population with a nominal size of 90 nm (Cat. 64009, size range 85.0 nm–94.0 nm) supplied by Polysciences (Niles, IL, USA) and is NIST-traceable. The sample consists also of a second mode at roughly 25 nm. Only the 90 nm mode is considered in this paper. The fourth sample is not a reference material but a commercial nanogold from BBI Solutions Ltd. (Nottingham, UK)with a single mode at roughly 50 nm.

Before their sizes can be measured using a Scanning Electron Microscope, the NPs are first deposited onto a silicon substrate (Agar Scientific AGG3390, Stansted, UK) with a spin coater following the protocol described in [5]. The substrate is functionalized with PLL (Poly-L-Lysine) to enhance the adhesion of the NPs [18,19] (See Figure 1).

### 2.2. Metrological Characterization of the Instrument

At the LNE, NP size measurements are carried out using a Zeiss Ultra-Plus Scanning Electron Microscope equipped with a Field Emission Gun (FEG) and a Gemini In-Lens SE detector (Zeiss, Oberkochen, Germany). This detector is located just above the sample at the end of the column. In contrast to standard chamber SE detectors, the In-Lens detector captures SE close to the sample surface making it particularly well suited for measuring the properties of nanoparticles. Once the instrument is installed, the first step in obtaining reliable results is to test the capabilities and limits of the SEM. This is necessary when determining the uncertainties of SEM measurements (see below) linked to the sample, instrument or environment, for instance. First, the SEM specific performance capabilities must be identified.

According to ISO standard 19749:2021 [20], the spatial resolution which is linked to a microscope’s ability to focus the electron in the beam must be determined per microscope. The contours of a particle can be accurately determined if the spatial resolution is small as they can be more easily distinguished on an image. In the Zeiss Scanning Electron Microscope installed at the LNE, the resolution was determined using the PATE 5.2.0 software and our standard use analysis conditions (3 kV accelerating voltage and 3 mm Working Distance, WD). The value in our instrument is (1.8 ± 0.3) nm (*k* = 1) which is very close to the value stated by Zeiss (1.7 nm at 1 kV).

The drift of the sample stage was also noted at 1 and 10 min [21]. Image acquisition is around 30 s, so stage drift is not considered in the final uncertainty.

Another parameter mentioned pertinent to the ISO standard is the cleanliness of the sample. The question is whether this parameter is inherent to the instrument and not dependent on the method, but SEM users must determine if this variable is instrument-induced or related to sample preparation and handling. The phenomenon of contamination disturbing the image acquisition induced by the e-beam exists in any Scanning Electron Microscope. The carbon molecules present on the substrate are attracted to the surface scanned by the beam and cover the particles to be analysed. A detailed study reporting more information can be found in ref. [21]. A number of precautions to avoid contamination have been suggested in the literature, such as pumping–holding the sample under vacuum for 12 h before taking measurements for analysis, or making adjustments to the electron column such as focusing outside the zone of interest. In this case, the effect of contamination does not represent a significant source of uncertainty.

Microscope noise was evaluated by using reference NPs (ERM FD-101b) deposited on Si. The images were analysed with the Mountains Lab v9.0.9789 (Digital Surf) software. The signal-to-noise ratio is defined by the standard as:(1)$$R_{SN}=\frac{A_{sig}}{\sigma_{sig}}$$

*A*_sig_ denotes the average value of the brightest or darkest parts of the image that show particles and *σ*_sig_ is the standard deviation of the signal. The ISO standard 19749:2021 [20] recommends *R*_SN_ ranges between 5 and 7. However, the *R*_SN_ value of our microscope was assessed at (30.9 ± 1.9) (*k* = 1) based on a series of 10 images.

The instability of the probe current can be another source of uncertainty. Stability measurements were carried out over several durations, short with 1 measurement of current every 10 s (11 measurements in total) and longer with 20 min analyses (1 measurement every minute). The relative standard deviation over a short period (10 s) was found to be only 0.3%, whereas over a typical analysis period it can reach 1.3%. Given the high signal-to-noise ratio in SEM images, we consider the probe current sufficiently stable for repeatable and reproducible SEM images in our instrument.

Finally, the most important stage in the qualification of the instrument is the magnification calibration, which includes a linearity study in both the horizontal and vertical directions. In our laboratory, the traceability of measurements is ensured by using a transfer standard, P900H60, designed by the LNE and produced by the Centre for Nanoscience and Nanotechnology (C2N, CNRS/Université de Paris-Saclay, Palaiseau, France) [22]. This reference structure is a 3D grating with a 900 nm nominal pitch along the XY axes (see Figure 2). It is composed of a succession of lithographed patterns with an average depth of 60 nm and an average grating pitch of 900 nm. An electron beam lithography system (Raith-Vistec EBPG 5000+) was used to fabricate the PMMA masks. The reactive ion etching technique was then applied to the masks to produce the gratings.

The 3D standard, P900H60, was calibrated by using the LNE’s metrological AFM [22]. This national reference instrument (Figure 2) took ten years to develop and is equipped with four laser differential interferometers in an original Abbe-compliant arrangement to minimize at maximum the measurement uncertainties [11]. A full uncertainty budget was established related to the mAFM metrology loop involved in the XYZ positioning of the tip relative to the sample [23].

A second calibration sample, (S1932B80, Agar Scientific, Stansted, UK), is used for low magnification calibration. This 2D standard grating, supplied by Agar Scientific has a pitch of 10 µm and is etched on a silicon chip with lateral dimensions of 5 mm × 5 mm and a thickness of 0.5 mm. SEM images are shown in Figure 2.

By using the P900H60 and S1932B80 transfer gratings together, linearity can be verified over the full magnification range in both the X and Y directions. As detailed in Section 3.1, the magnification measurements are performed at three different magnifications associated with three different gains.

### 2.3. Uncertainty Budget

Nanoparticle size measurements based on SEM images are traceable to SI units if the microscope calibration is linked with the size measurement uncertainties. First, we identify the main sources of error linked to the size measurement procedure of the silica NPs by following the Ishikawa’s approach of the five Ms, i.e., taking into account Material, Medium, Method, Machine and Man power [24]. Each source of uncertainty was evaluated and the budget was published in ref. [21]. These sources of uncertainty include repeatability measurements, magnification, beam width, operator, pixel size, image analysis, contrast, brightness, scan speed, drifts, noise reduction parameter, focus and astigmatism adjustments. Some of these are considered negligible and are not covered in this article (scan speed, contrast, brightness, focus and astigmatism adjustment). When considering the six main contributions to uncertainty (pixel size, repeatability, magnification, beam width, threshold selection during image treatment and manpower) the final uncertainty is 10% (*k* = 1) for the silica NP size measurements. A simplified uncertainty budget is described in Section 3.1 to help generalise the approach to all particles.

### 2.4. Image Analysis

Microscopy image and data analysis processing is also a key step of the measurement procedure. The NP dimensions are deduced from the edge-to-edge distance measurements made on the NP image using specified software. This data-processing software requires, first of all, that the edges of the particle are clearly distinguishable. However, the accuracy of the measurement depends on contour definitions in the data-processing software. Most commercially available algorithms are black boxes and information on the calculation of results is not provided to user. At the LNE we developed our own software platform, Platypus (v.2.2.45), in collaboration with the French SME Pollen Metrology (Grenoble, France). The measurement of NP size is always carried out at the Full Width at Half Maximum (FWHM) of a profile defined between both edges of the particle. In Section 4, this question about thresholding is discussed.

## 3. Results and Discussion

### 3.1. Calibration of the Magnification

SEM metrology is driven by the identification of two elements in a digitalized image and the determination of the distance between them. Since the image consists entirely of pixels, their size is thus a crucial element upon which the entire SEM metrology is based. The measurement unit is defined as the scan width divided by the number of pixels in the measurement system. Therefore, the goal of Scanning Electron Microscope calibration is to first calibrate the pixel size in the X and Y scanning directions. In other words, the aim of magnification calibration, and so, pixel size calibration, is to correctly convert pixels into SI length units. The use of 2D gratings with nanoscale lateral periodic patterns facilitates this conversion. The calibration procedure consists of measuring the pitch corresponding to the distance between two similar edges (Figure 2) with an mAFM and a Scanning Electron Microscope. The results of the mAFM measurements performed on P900H60 and S1932B80 are reported in Table 1.

Performing this procedure with two standards, P900H60 and S1932B80, enables the verification of linearity over the entire magnification range in the X and Y directions (Table 2). In our case, the full range is made up of three magnification ranges associated with three different gains.

After image acquisition, pitch is evaluated using the “Lattice and Lateral calibration” tool from the Mountains Lab^®^ (Digital Surf) software. This tool uses a Fast Fourier Transform (FFT) to automatically evaluate the pitch in the two directions. The standard ISO 11952:2019 [25] indicates that to correctly use this tool a minimum of seven periods (ideally 10) have to be visible on the image. As the smallest pitch available at the laboratory is around 900 nm, the highest magnification that we can calibrate is 25 kX. Beyond this magnification, another strategy has to be implemented, as in the case with NPs.

Figure 3 shows an example of measurements taken over the entire range of magnifications along the X-axis.

For each nominal magnification, the actual magnification is recalculated by determining the mean pitch:(2)$$G_{actual}=\frac{G_{nominal}\cdot P_{meas.}}{P_{ref}}$$

With,
*G*_actual_, the actual magnification value.*G*_nominal_, the nominal magnification value.*P*_meas._, pitch measured from the SEM images of the standard.*P*_ref_, reference value of the pitch given by the mAFM.*q* denotes the correction factor linked to calibration and can be defined as:



(3)
$$q=\frac{G_{actual}}{G_{nominal}}=2-\frac{p_{actual}}{p_{nominal}}$$

*p*_actual_, the pixel size determined with the standard structure.*p_nominal_,* the nominal value of the pixel size.


In accordance with ISO 16700:2016 [26], an instrument can be declared compliant if a relative deviation of less than 5% (tolerance zone in Figure 3) is observed between the X and Y directions. An additional precaution can be taken. The instrument is also deemed to be compliant if a drift of less than a maximum allowed error (1.5% at the LNE) is observed in the X and Y directions compared with the last performed calibration. If the results are within the tolerance zone, a correction factor should be applied to all the measurements carried out following this calibration. This calibration procedure must be performed every three months or after each maintenance operation.

The images acquired with P900H60 allow us to assess the leading edge distortion. When scanning a new line, the beam is deflected to change direction leading to a variation in scan speed at the beginning of each line. If this deviation is significant, the first pixels have to be omitted for both calibrations and NP measurements. Therefore, the pitch between each pattern of a P900H60 sample was measured along the X-axis for a total of ten pitches. The results are shown in Figure 4. The non-linearity of the scan is observed and represents a relative difference of 2.2%. On average, for the left quarter of the image the relative deviation is 0.5% and becomes 0 for the left third of the image. Regarding nanoparticle size measurements, this effect can be considered negligible for the following reasons: (i) the effect will only be applied to a tiny portion of nanoparticles in the image and (ii) the relative deviation decreases rapidly, so this effect is infinitesimal compared with other sources of uncertainty.

### 3.2. Uncertainty Assessment Extended to All Particles

As mentioned above, a comprehensive study conducted on several silica NP populations allows us to establish an uncertainty budget [22]. In this section, we propose simplifying and extending this approach to all types of nanoparticles.

After calibrating the instrument, *D*_eq_, the ECD of a population composed of N particles must be corrected on the basis of the calibration procedure, image pixel size and imaging conditions. Then, *D*_eq_corr._, the ECD-corrected value is calculated as following:

If the pixel size is larger than the targeted uncertainty, a part of the two pixels that define the limits of the particle is counted two times,
(4)$$D_{eq\_corr.}=\left(D_{eq}-p_{actual}\right)\cdot q$$

If the pixel size is smaller than the targeted uncertainty,
(5)$$D_{eq\_corr.}=\left(D_{eq}\right)\cdot q$$

In the general, the pixel size is very often greater than the measurement uncertainty, so the compound variance of *D*_eq_ is calculated by applying the propagation law to Equation (3):(6)$$u_{corr.}^{2}\left(D_{eq_{corr}.}\right)={\left(D_{eq}-p_{actual}\right)}^{2}u_{corr.}^{2}\left(q\right)+q^{2}u_{corr.}^{2}\left(D_{eq}\right)+q^{2}u_{corr.}^{2}\left(p_{actual}\right)$$

#### 3.2.1. Uncertainty in the Determination of *D*_eq_

The uncertainty associated with the determination of *D*_eq_ is calculated as:(7)$$u_{corr.}^{2}\left(D_{eq}\right)=u_{bw}^{2}\left(D_{eq}\right)+u_{s}^{2}\left(D_{eq}\right)+u_{r}^{2}\left(D_{eq}\right)+u_{d}^{2}\left(D_{eq}\right)$$
and results from the combination of four independent uncertainties:electron beam width, $u_{bw}(D_{eq})$

The uncertainty associated with the width of the e-beam corresponds directly to the size of the beam. This uncertainty is evaluated in ref. [16]. Under conventional imaging conditions, with the beam size equal to (1.8 ± 0.3) nm (k = 1) (see Section 2.2), the variance due to beam size is equal to:(8)$$u_{bw}^{2}\left(D_{eq}\right)=1.8^{2}$$

Sampling, $u_{s}(D_{eq})$

The variance is related to the number of measured particles (see ISO 13322-1:2014 [6]) and is written as:(9)$$u_{s}^{2}\left(D_{eq}\right)=s_{s}^{2}$$

*s*_s_^2^ denotes the standard deviation calculated with the size measurements of the entire population.

Measurement repeatability, $u_{r}(D_{eq})$

The uncertainty due to the repeatability of size measurements is expressed by a dispersion characteristic.
(10)$$u_{r}^{2}\left(D_{eq}\right)=s_{r}^{2}$$

*s*_r_^2^ denotes the standard deviation calculated with repeated size measurements.

Instrument drifts, $u_{d}(D_{eq})$

Instrument drift is observed between two calibrations leading to a bias *b* in the size measurement. If we denote this bias in the measurement of lateral dimensions (>1.5%) by ± *b* and assume that it follows a uniform distribution, the variance associated with the drift is written as:(11)$$u_{d}^{2}\left(D\right)={\left(D_{eq}\frac{b}{\sqrt{3}}\right)}^{2}$$

#### 3.2.2. Uncertainty Associated with Pixel Size

When the pixel size is larger than the beam size, the uncertainty associated with the pixel size becomes predominant. It then follows a uniform distribution:(12)$$u_{corr.}^{2}\left(p_{actual}\right)={\left(\frac{p_{actual}}{\sqrt{3}}\right)}^{2}$$

#### 3.2.3. Uncertainty Associated with the Correction Relative to Calibration

The uncertainty in the determination of the correction factor, *q*, results from the combination of the uncertainties arising from the uncertainty in the mean pitch of the mean grating pitch given by the calibration certificate for the standard structure, $u_{ref}\left(P_{ref}\right)$, and the measured grating pitch, $u_{meas.}\left(P_{meas.}\right)$

From the relationships given in (2) and (3), *q* can be expressed as:$$q=\frac{P_{meas.}}{P_{ref}}$$

As a result, the compound variance of *q* is determined by applying the law of variance propagation:(13)$$u_{corr.}^{2}\left(q\right)={\left(\frac{1}{P_{ref}}\right)}^{2}{u_{meas.}}^{2}\left(P_{meas.}\right)+{\left(\frac{P_{meas.}}{P_{ref}^{2}}\right)}^{2}{u_{ref}}^{2}\left(P_{ref}\right)$$

Grating pitch calibration

The uncertainty associated with the calibration of the microscope is taken from the uncertainty shown on the calibration certificate for the reference material.
(14)$${u_{ref}}^{2}\left(P_{ref}\right)=u_{ref\_certificate}^{2}$$

Grating pitch measurement

The uncertainty associated with measuring the grating pitch is expressed as a standard deviation calculated from several measurements taken at different points on the reference structure.
(15)$${u_{meas.}}^{2}\left(P_{meas.}\right)=s_{L\_meas.}^{2}$$

#### 3.2.4. Variance Expression Linked to the Corrected Diameter Determination

The uncertainty associated with the corrected value of the equivalent circular diameter, $u_{corr.}^{2}\left(D_{eq\_corr.}\right)$, is given by Equation (6).

From Equations (12)–(15), we can write:(16)$$u_{corr.}^{2}\left(D_{eq\_corr.}\right)={\left(D_{eq}-p_{actual}\right)}^{2}\left({\left(\frac{1}{P_{ref}}\right)}^{2}s_{L\_meas.}^{2}+{\left(\frac{P_{meas.}}{P_{ref}^{2}}\right)}^{2}u_{ref\_certificate}^{2}\right)\;+q^{2}\left({\left(1.8\right)}^{2}+s_{s}^{2}+s_{r}^{2}+{\left(D_{eq}\frac{b}{\sqrt{3}}\right)}^{2}+{\left(\frac{p_{actual}}{\sqrt{3}}\right)}^{2}\right)$$

### 3.3. Accelerating Voltage: A Determining Parameter

Measuring the dimensions of a nanoparticle imaged by SEM first involves detecting its edges. Theoretically, the size can then be determined from a profile extracted along the line corresponding to the particle diameter. However, as observed in Figure 5, the profile edges are not perfectly vertical. Consequently, the question arises as to exactly where the measurement should be taken. The signal intensity at this boundary is called the threshold signal level (TSL). Since the early years of sub-microstructure measurements in a Scanning Electron Microscope [27], it has been conventionally accepted that the TSL should be placed at the half-signal as shown in Figure 5. Platypus (v.2.2.45), the software used for an automated image analysis, has been programmed to perform NP binarized segmentation at Full Width at Half-Maximum (FWHM).

In a previous study, we demonstrated that the NP size measurement depends on the e-beam acceleration voltage (AV or landing energy) applied by the operator by adjusting the parameter often called EHT (Electron High Tension) [15]. It should be noted that the landing energy and acceleration voltage can be slightly different if the material is non-conductive, leading to charging if deceleration is applied to the e-beam. Figure 6 shows the variations in measured NP size between 25 nm and 90 nm and chemical properties (gold, silica and PSL) when EHT varies between 1 and 15 kV. The measured size differs with EHT but the curves show similar trends. The values first decreases in the low voltage range (EHT < 2–3 kV) and reaches a minimum. Then, the curve increases (silica and PSL) or plateaus (gold).

In fact, the changes observed in Figure 6 can be explained by a dependence of the TSL on the size and chemical composition of the nanomaterial. This phenomenon has already been observed in a Scanning Electron Microscope operating in transmission mode (TSEM, Transmission Scanning Electron Microscopy) [13]. It is explained by edge effects and by the different scattering properties of the analysed NP. In fact, electrons scatter differently in materials with high atomic numbers, such as gold nanoparticles, than in a material with a low density, such as polystyrene.

In SEM, the collection of SEs is highly dependent on two determining physical parameters: the maximum penetration depth, λ_MPD_ and the escape depth, λ_ED_.

λ_MPD_ of primary electrons is defined as the maximum depth at which they slow down to rest. If the e-beam is perpendicular to the sample surface, λ_MPD_ depends on two parameters: EHT and the atomic number of the material being studied. The primary electrons of the e-beam penetrate the material with an energy corresponding to EHT and then scatter into the nanomaterial. In reality, primary electrons interact with atoms and are scattered elastically or inelastically. In the latter case, they give up part of their energy to the system, which then relaxes by especially emitting SEs. All along the path of the primary electrons, this progressive loss of energy with each collision results in the electrons slowing down completely. Only a fraction of the generated SEs will be able to move towards the surface, escape the sample and be collected by the detector. λ_ED_ corresponds to the minimum energy required for the electrons to reach the surface and escape from the material.

It is well known that the secondary electron yield (SEY), defined as the ratio of emitted SEs to incident primary electrons, reaches a maximum when λ_MPD_ is equal to λ_ED_ [28]. It had been previously shown that the minimum values of the measured sizes, reported in Figure 6, corresponded to the maximum signal-to-noise ratio observed in profiles (example in Figure 5) and so are related to the maximum SEY [15]. It has been noticed that here the signal-to-noise ratio is not calculated from Equation (1), but estimated only as the ratio of nanoparticle signals to the average of the substrate signals.

Moreover, we have demonstrated that this minimum value of the size measurement matches with the true size of the NP by comparing SEM and TEM results obtained from reference nanomaterials. The impact of EHT on the resulting histogram of size distribution is shown in Figure 7.

The variations in the size histograms as a function of EHT are greater for the largest silica particles (80 nm). A discrepancy of roughly 12 nm is observed between 1 kV and 3 kV. For silica NPs with 25 nm size, only 2 nm separates the measurements performed at 2.5 kV and 8 kV. Furthermore, in the complete range of EHT, the measurements are within the uncertainty. If we compare similarly sized PSL and silica particles, the dispersions of measurements are very different. The measured dimensions vary by only 2 nm. For the gold particles, the size distribution shifts towards small values with a maximum of 4 nm over the entire energy range, without reaching a minimum.

The Figure 8 depicts SEM images of different silica NPs with 26 nm (FD-304) and 80 nm (FD-101b) sizes at different EHT ranging from 1 to 12 kV associated with their profiles measured along their diameter. At 1 kV, the central part of the particle is dark, as seen especially for the 80 nm NPs. The primary electrons penetrate the centre of the NP with very low energy and after scattering inside the particle, very few of them have enough energy to escape from the particle. Most of the electrons are absorbed by the material and are discharged via the ground connection. In contrast, at the edges, the trajectory of primary electrons remains close to the edge, which facilitates the escape of SEs and increases the possibility of their collection by the detector [15]. As a result, the edges appear brighter. The dark region is less pronounced for the smallest particles (26 nm) because the edge effects represent a large part of the particle. Around 8 kV, the signal-to-noise ratio decreases sharply and for the smallest particles (D < 50 nm), edge detection becomes difficult. From a certain accelerating voltage, λ_MPD_ of high-energy primary electrons increases dramatically and they penetrate too deeply into the particle. Consequently, the energy left over from multiple collisions with atoms is not enough for them to reach to the surface of the particle. At higher energies, they are even able to pass through the surface. So, as the energy increases, the detector collects fewer and fewer SEs and the signal-to-noise ratio falls significantly [15]. In contrast, for EHT close to 3 kV, the signal-to-noise is at its maximal value and the NPs appear brighter without a dark core. As mentioned above, at this optimum energy value, λ_MPD_ and λ_ED_ are equal.

As seen in Figure 9a, the behaviour described above is slightly different for gold NPs, which have a higher density than silica. Within a material with a larger atomic number and at comparable energy levels (constant EHT), the average free path is shorter and the probability of collisions with gold atoms is greater. In the range of low EHT (1–3 kV), the central part of the NP is darker with a small dip in the middle of the profile. This dip gradually disappears as the energy of the primary electrons increases. From 6 kV to 15 kV, the signal-to-noise ratio seems to be constant and the curve seen in Figure 6 reaches a plateau. However, above 15 kV, the signal-to-noise ratio will eventually fall as the number of electrons able to pass through the particle increases significantly. Moreover, increasing the EHT too much could damage the particles and make it impossible to measure them.

However, the behaviour of PSL NPs (Figure 9b) is similar to that of the silica reference nanomaterials (Figure 8). The dip observed in the middle of the particle disappears temporarily at around 2 kV. Below 2kV, the dark central mark clearly visible is explained by the penetration of primary electrons into the NP at too low an energy to generate SEs that are able to escape from the particle. Conversely, the dark central part reappears above 8 kV because the high-energy primary electrons penetrate too deeply into the particle and few of the generated SEs have enough energy to return to the surface.

However, unlike STEM mode, NP dimensional metrology based on the collection of secondary electrons is also highly dependent on the nature of the substrate on which the nanoparticles are deposited. In fact, the signal from the edges of the particle is the result of a convolution between the SEs generated by the substrate and the particles. In contrast to Figure 8 and Figure 9, which show different NP groups of the same nature and size, Figure 10a shows exactly the same single FD-101b particle of 75 nm, imaged at several acceleration voltages and deposited on a silicon substrate. During these measurements, the contrast and luminosity were kept constant over time so that the measurements could be compared. The dark mark at the centre of the particle described above appears at 5 kV and becomes more intense as the acceleration voltage increases.

Higher than 12 kV, the contrast and brightness are no longer adequate and the particle is difficult to observe. In the measurement profiles (Figure 10b), from 3 kV onwards, the signal widens and the central part deepens, while the total signal intensity decreases as the voltage increases. In Figure 10c, the background level corresponding to the silicon surface signal and the signal-to-noise ratio of the particle relative to the silicon substrate are plotted on a graph. We can see that the signal-to-noise ratio peaks at around 3 kV, while maintaining a minimal background signal. The background signal increases and reaches a plateau with an EHT between 5 and 9 kV. Over this voltage range, the profile widens and reaches its maxima before decreasing sharply. Additional details regarding the edge effects measured on the 75 nm FD-101b NP are included in Figure 11a. The signal has been normalised to better observe the widening of the signal as a function of EHT. This broadening of the signal profile is due to a convolution effect between the signal coming from the NP and the substrate.

To further understand the impact of the nature of the substrate on the measured profiles, a single 82 nm FD-101b nanoparticle was deposited on a carbon film and imaged at several voltages between 3 kV and 7 kV in Figure 11b. The evolution of the profiles for a particle deposited on silicon or carbon is very different. These results show the importance in SEM metrology of the substrate on which NPs are deposited. The edge effects observed with silicon substrates are not observed for carbon substrates. The signal-to-noise ratio for particles deposited on carbon reaches a maximum at around 3 kV but no signal broadening is observed. The edge effects observed in Figure 11a can effectively be explained by the convolution of the signals from the NP/silicone substrate and multiple interactions with SEs. The secondary electrons generated by the silicone substrate and escaping from the surface can penetrate the NP and interact with its atoms to generate new SEs.

Finally, this study shows that the profile shape and the SE yield changes depending on the EHT of the electron beam. These changes to the profile shape demonstrate that NP size cannot always be determined from a threshold at FWHM, the current method used by our Platypus software. Research at the Physikalisch-Technische Bundesanstalt (PTB), the German Institute of Metrology, observed a link between the measurements associated with nanoparticles and the chemical composition of the analysed sample [16]. The TSL depends on the EHT and the nature of the substrate on which the nanoparticles are deposited. In transmission mode, a model was developed by the PTB for automatic threshold determination and reliable NP segmentation as a function of the various parameters. The collection of SEs in SEM mode has also been modelled by the LNE and includes the effect of EHT as well as the properties of the studied nanomaterial (geometry, density, chemical composition, etc.) and substrate. The interactions between electrons and the nanomaterial as they pertain to the SE yield was modelled using JMONSEL (Java Monte Carlo Simulation of Secondary Electron) [15,29].

An operator who wants to reliably measure the size of nanoparticles in SEM and who does not have access to the model described above can carry out measurements on the nanoparticles by varying the acceleration voltage over a range of 1 to 15 kV and determine the value of the voltage corresponding to the minimum value of the particle diameter as shown in this section. For instance, in the case of silica particles, this minimum value of NP diameter corresponds to the maximum value of the signal-to-noise ratio and the minimum value of the substrate signal (Figure 10).

### 3.4. Diagram for a New Scanning Electron Microscope Calibration Process

The diagram in Figure 12 shows the five steps required to generate reliable and traceable NP size measurements: calibration/monitoring, Scanning Electron Microscope qualification and optimum acceleration voltage (EHT) setting. The Scanning Electron Microscope calibration procedure can be performed with a commercial transfer standard supplied with a calibration certificate (step 02). But, this reference structure must be periodically checked at least once every two years (step 01). The calibration will have to be carried out by optical/X-ray interferometry or by means of a metrological AFM by a National Metrology Institute (INM). The transfer standard’s reference values are important for insuring the calibration of Scanning Electron Microscope pixel size and magnification. This Scanning Electron Microscope calibration procedure should be performed periodically; our suggestion is once every four months as well as before and after microscope maintenance. Thereafter, the period between two calibrations may vary according to the measured deviations. An equipment monitoring procedure must also be carried out on a regular basis (step 05). For this, measurements of reference nanoparticles (control standard) with assigned quantity values must be performed monthly. The purpose of this monitoring procedure is to check the trueness of the measurements performed with Scanning Electron Microscopes, highlighting instrument drift and verifying the stability of the entire measurement procedure.

Assessing uncertainties is a lengthy process, but it is necessary to evaluate confidence in a measurement. Type A and B uncertainties must be evaluated (step 03) according to the principles of the GUM (Guide to the Expression of Uncertainty of Measurement) and guidelines as summarized in this document [30]. This work must be applied to each type and size of nanomaterial, as well as to any SEM instrument.

In this paper, we also want to highlight the impact of EHT on the reliability of NP size measurements (step 04). In general, experienced SEM users optimize the accelerating voltage based on the desired microscope magnification. In this study, we demonstrate that the SEM signal processing used to determine the size of the nanoparticle is critical and depends on the EHT value, as well as the size and the chemical composition of the particle. One solution might be to use a reference nanomaterial to choose the EHT parameter, but this is unrealistic, because we would need nanoparticles of varying compositions and sizes. Furthermore, very few certified reference nanomaterials are currently available on the market. A list of reference nanomaterials commercially available worldwide, together with links to manufacturers/providers, has been compiled by ISO TC229 (see for example the list of currently available nanoscale reference materials provided by BAM: www.nano-refmat.bam.de, webpage consulted on 31 January 2024). An alternative method for improving the reliability of the size measurements is to incorporate a model into the automatic algorithms that considers the properties of the particles, the acceleration voltage and the nature of the substrate. Finally, we propose a third and simpler approach which requires pre-experimental studies relating to the role of EHT on size measurements. Depending on the curve obtained by these measurements, the optimum value can be determined and SEM users can adjust their EHT parameter appropriately.

## 4. Conclusions

This article proposes a comprehensive methodology for ensuring the traceability of nanoparticle size measurements in SEM. Usually, the dimensional metrology of nanoparticles by SEM focuses on particles larger than 10 nm. The microscope calibration procedure, which involves calibrating pixel size by using transfer standards and utilising a metrological AFM, is described in detail. This first step is aimed at reaching accurate measurements. This paper also contains guidelines for estimating the uncertainty associated with NP dimensional measurements in SEM. Each uncertainty source was evaluated and the instrument was metrologically qualified in accordance with ISO standard 19749:2021 [20]. This study also highlights the fact that the calibration process is not sufficient for accurate and reliable results. Particular attention must be paid to role of accelerating voltage, EHT, in size measurements. We show that the relationship between size and acceleration voltage depends on the chemical nature of the particles as well as their sizes. But, there will never be enough certified reference nanomaterials (CRM) to adjust the correct value of the acceleration voltage to accurately reach the size of any NP populations. Therefore, two alternative methods are proposed here to significantly improve the reliability of SEM measurement results. The first option is to simulate the line-scan profile of a nanoparticle imaged at any EHT. The LNE has developed a specific model capable of finding the correct TSL for an accurate measurements of NP size. The second method consists in plotting the dependence of size on EHT. A universal curve has been identified and all the nanoparticles studied by the LNE exhibit a minimum or a plateau. We have demonstrated that adjusting the EHT parameter based on this minimum/plateau leads to accurate results. Furthermore, the nature of the substrate on which the nanoparticles are deposited has an impact on the shape of the SEM signal and must considered when determining NP size. Finally, the various steps involved in establishing traceability are summarised in a diagram.

## Figures and Tables

**Figure 1 nanomaterials-14-00931-f001:**
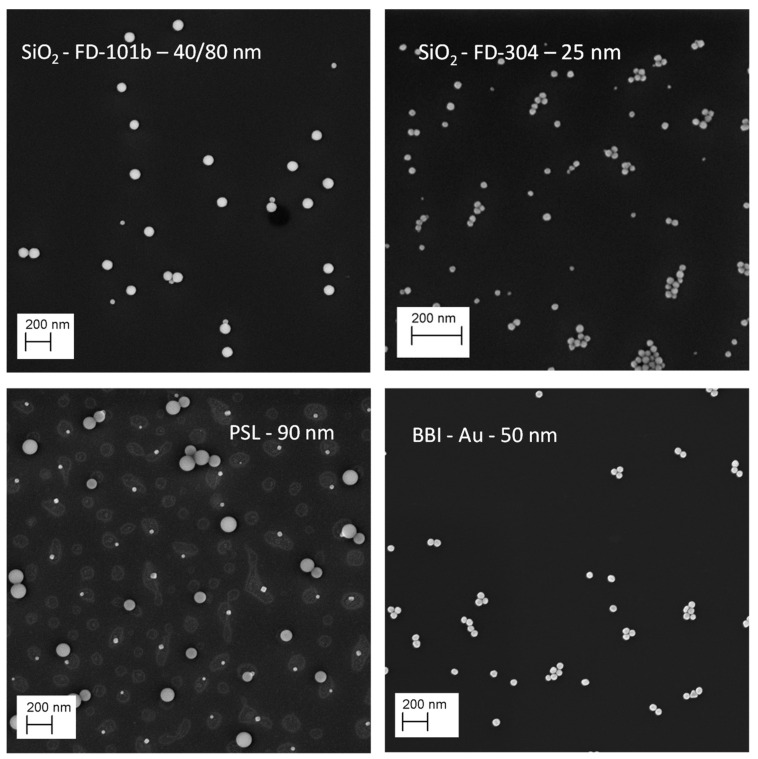
SEM images of the four studied samples in this study: FD-101 b 40/80 nm bimodal nanosilica (×20,000), FD-304 25 nm monomodal nanosilica (×40,000), bimodal PSL(×20,000), only the 90 nm population is considered here, and BBI 50 nm nanogold (×20,000).

**Figure 2 nanomaterials-14-00931-f002:**
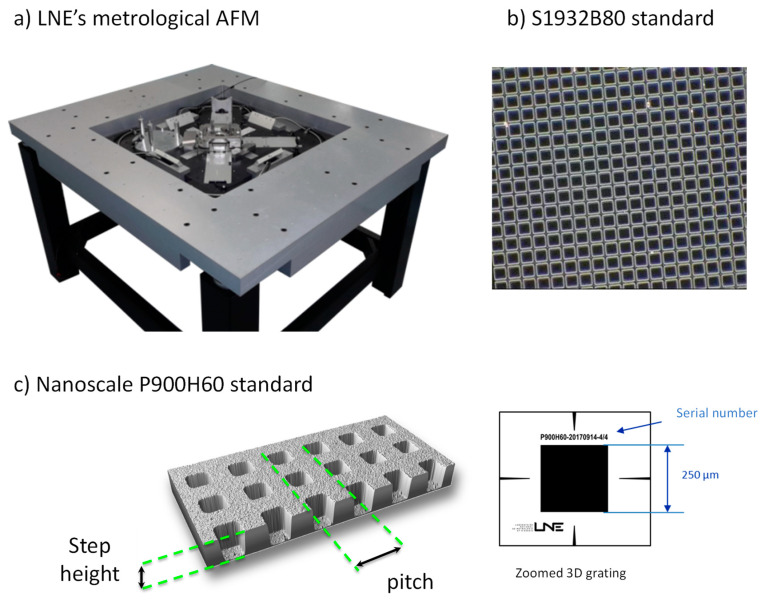
(**a**) LNE’s metrological AFM, the national reference instrument and keystone in the traceability chain for nanometre-scale measurements; (**b**) 2D standard grating (S1932B80) with a pitch of 10 µm, etched on a silicon chip; and (**c**) 3D pattern grating P900H60 developed in collaboration with CNRS/C2N. The grating pitch (along the X and Y axes) is equal to (899.9 ± 2.0) nm with a step height equal to (68.2 ± 1.0) nm.

**Figure 3 nanomaterials-14-00931-f003:**
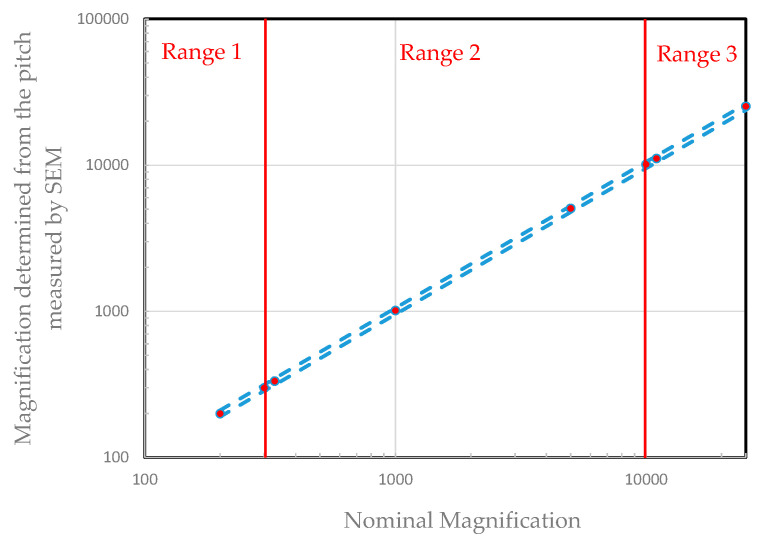
Example of a graph demonstrating the linearity of grating pitch measurements over the entire X-axis magnification range. The blue dotted lines indicate the tolerance zone (±5%). Every red dots, corresponding to measurement points, are within this tolerance zone for the three ranges.

**Figure 4 nanomaterials-14-00931-f004:**
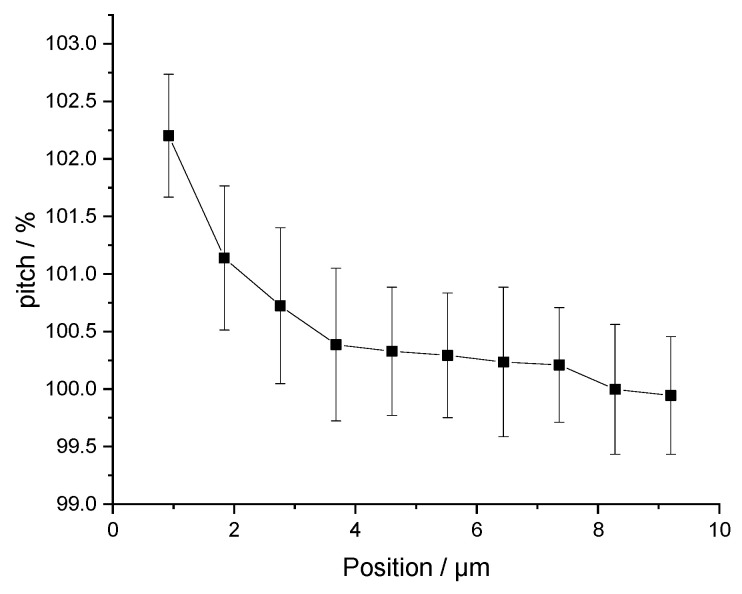
Pixel size deviation observed on the left part of a P900H60 sample image acquired in a Scanning Electron Microscope due to leading edge distortion. The pitch is measured at different positions along the X-axis.

**Figure 5 nanomaterials-14-00931-f005:**
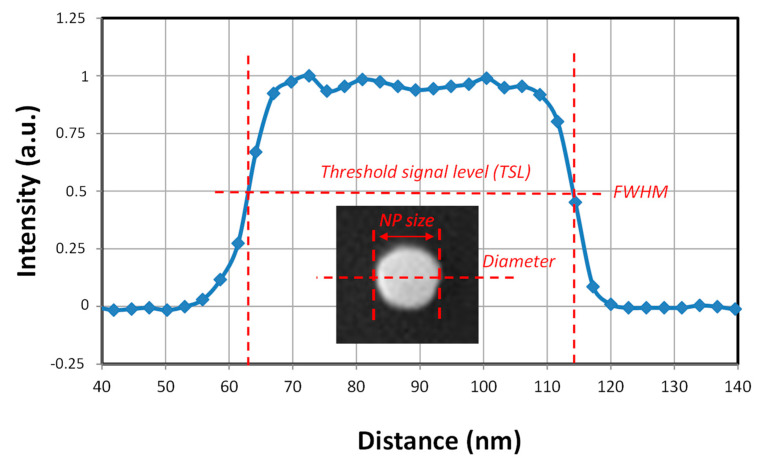
Profile extracted from SEM image of a single gold NP and performed along its diameter. TSL at Full Width at Half-Maximum (FWHM) are indicated in the Figure.

**Figure 6 nanomaterials-14-00931-f006:**
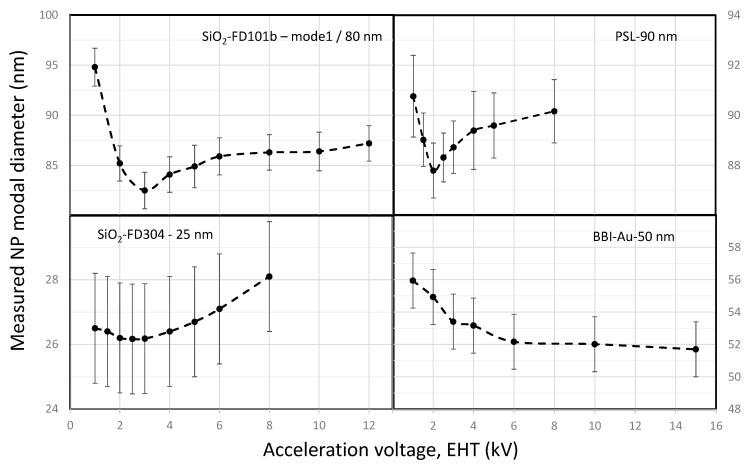
EHT dependence of the NP modal diameters measured on various nanomaterials: 80 nm FD-101b, 25 nm FD-304 and 90 nm PSL nanoparticles and 50 nm nanogold from BBI.

**Figure 7 nanomaterials-14-00931-f007:**
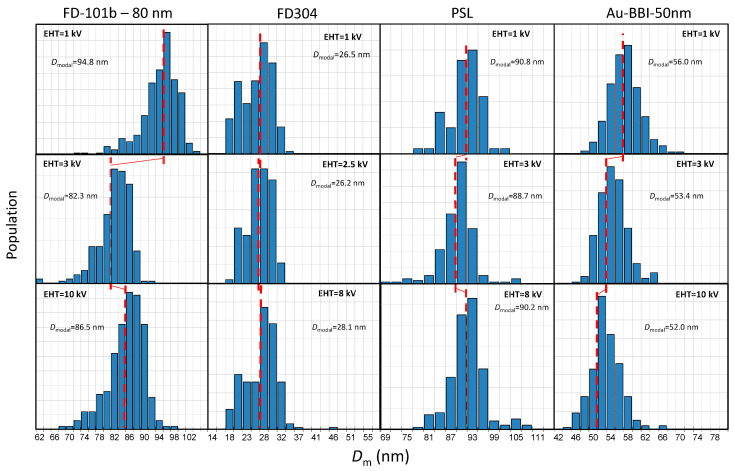
Histograms showing the size distribution of the four studied samples (FD-101b, FD-304, PSL and BBI nanogold) compiled from measurements performed at different acceleration voltages (EHT) ranging from 1 kV to 10 kV. The value of the modal diameter is given for each histogram to highlight the impact of the EHT parameter on the measured size.

**Figure 8 nanomaterials-14-00931-f008:**
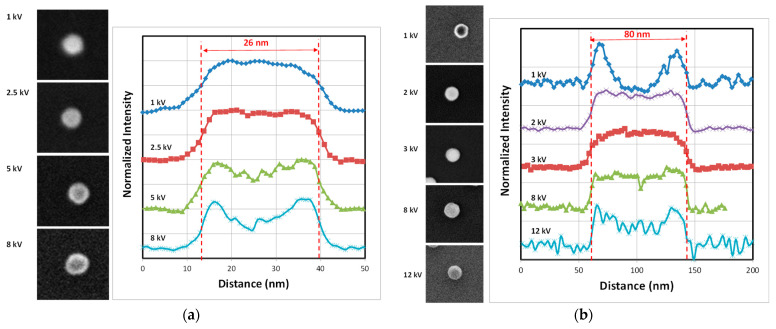
SEM images of 26 nm FD-304 (×40,000) (**a**) and 80 nm FD-101b (×20,000) and (**b**) silica nanoparticles at different EHT ranging from 1 kV to 12 kV. Profiles of each nanoparticle are measured along the NP diameter and are displayed to the right of the SEM images.

**Figure 9 nanomaterials-14-00931-f009:**
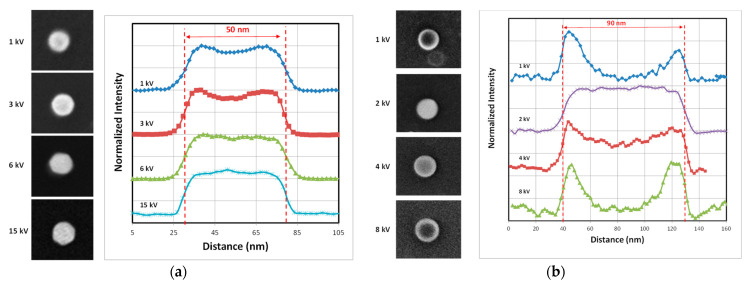
SEM images of 50 nm gold nanoparticles (×20,000) (**a**) and 90 nm PSL nanoparticles (×20,000) (**b**) at different EHT ranging from 1 kV to 15 kV. Profiles of each nanoparticle measured along the longest part of the NP are seen to the right of the SEM images.

**Figure 10 nanomaterials-14-00931-f010:**
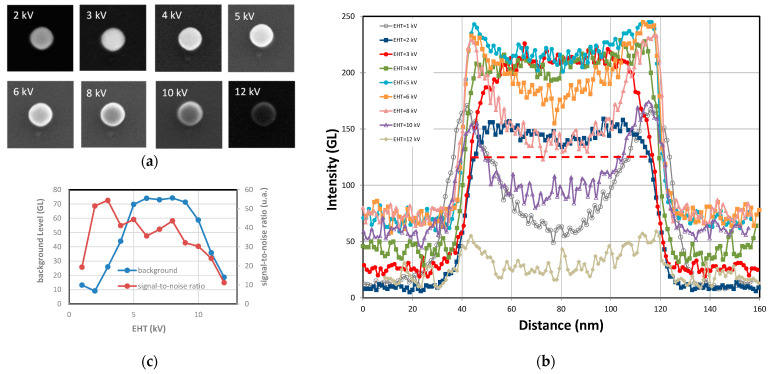
(**a**) SEM images of a single 75 nm FD-101b particle (×50,000) deposited on Si-wafer and acquired at different EHT; (**b**) line-scan profiles associated with these images at various EHT; and (**c**) intensity of the silicon background and signal-to-noise ratio of the profiles as a function of EHT.

**Figure 11 nanomaterials-14-00931-f011:**
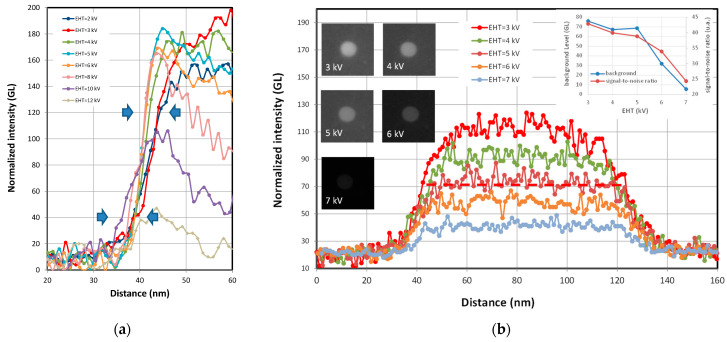
(**a**) Zoomed-in view on the edges of the normalized profiles reported in Figure 10 and acquired on a 75 nm FD-101b NP (×50,000) deposited on silicon substrate; (**b**) profiles of a single 82 nm FD-101b particle deposited on carbon substrate (copper grid) and imaged at various accelerating voltages; (left insert) SEM images of an NP at various EHT; and (right insert) background level and signal-to-noise ratio as a function of EHT.

**Figure 12 nanomaterials-14-00931-f012:**
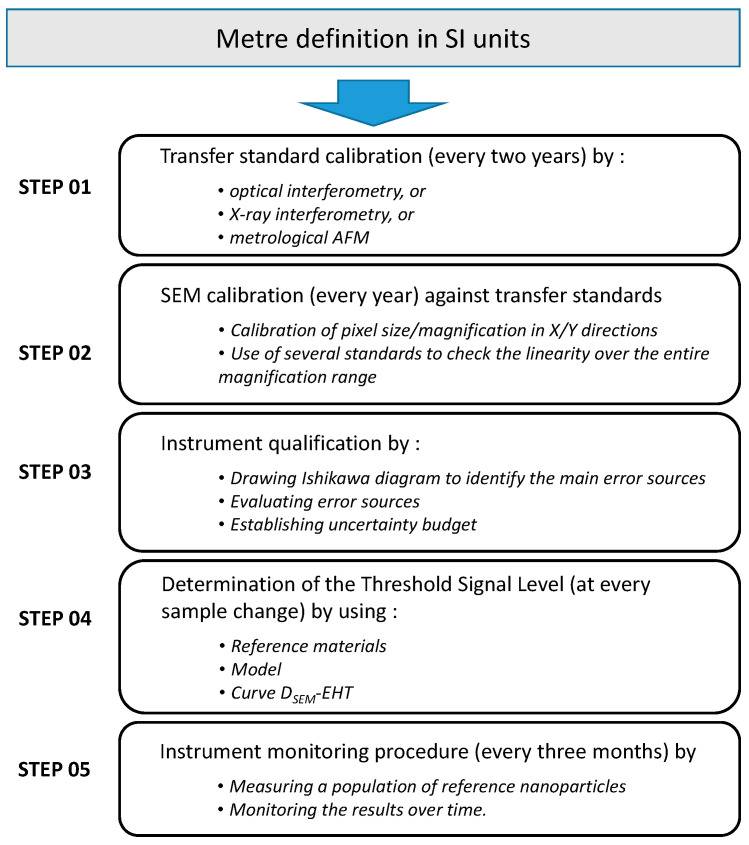
Diagram summarizing the various steps required to provide NP size measurements traceable to SI units with calibration, instrument qualification and TSL determination.

**Table 1 nanomaterials-14-00931-t001:** mAFM measurement (*P*_reference value_) carried out with the two standard structures: P900H60 and S1932B80.

Standard Structure	Pitch Measurement Associated with Uncertainty (*k* = 2)
P900H60	(899.9 ± 2.0) nm
S1932B80	(10,002 ± 10.0) nm

**Table 2 nanomaterials-14-00931-t002:** Magnifications to be evaluated and appropriate standards to check linearity over all the ranges.

Range	Magnification	Standard to Be Used
1	×200	S1932B80
	×300	S1932B80
2	×330	S1932B80
	×1000	S1932B80
	×5000	P900H60
	×10,000	P900H60
3	×11,000	P900H60
	×25,000	P900H60

## Data Availability

Data are contained within the article.

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
