# Peer review of "Strategy for Ensuring the Metrological Traceability of Nanoparticle Size Measurements by SEM"

_nanomaterials, 2024, doi:10.3390/nano14110931_

Round 1

Reviewer 1 Report

Comments and Suggestions for Authors

The work entitled “Strategy for ensuring the metrological traceability of nanoparticle size measurements by SEM” describes the calibration procedure set up at LNE (Laboratoire National de métrologie et d’Essais) and proposes a general evaluation method of the uncertainties for measuring nanoparticle size by SEM. A specific study is presented here to investigate the impact of the energy of the primary electrons(PE) generated by e-beam or accelerating voltage (EHT) on the reliability of the size measurements. This research presented a diagram describing the various stages involved in establishing the traceability for the SEM measurements of nanoparticle size to facilitate the work of future operators. But before the reviewer recommends this work to be published in nanomaterials, some issues need to be tackled.

1.        The full text lacks innovation in scientific paper writing, and the experimental characteristics are more similar to patents.

2.        As shown in Figure 1, to highlight the differences in NP behaviour regarding their size and chemical composition during electron/matter interactions induced by the SEM e-beam, four samples of metal, oxide, and polymer nanoparticles were studied. So, why did you choose these four samples for study?

3.        As shown in Figure 6, when the acceleration voltage of gold nanomaterials is greater than 6 kv, the modal diameter tends to become a constant value. What are the specific reasons accounting for this result?

4.        Where are relations (16)(17)(18) and (19)?

5.        For materials with poor radiation resistance, how to use this method to accurately measure distances?

6.        How to avoid the influence of current in scanning electron microscopes?

Comments on the Quality of English Language

It can be improved.

Author Response

The work entitled “Strategy for ensuring the metrological traceability of nanoparticle size measurements by SEM” describes the calibration procedure set up at LNE (Laboratoire National de métrologie et d’Essais) and proposes a general evaluation method of the uncertainties for measuring nanoparticle size by SEM. A specific study is presented here to investigate the impact of the energy of the primary electrons (PE) generated by e-beam or accelerating voltage (EHT) on the reliability of the size measurements. This research presented a diagram describing the various stages involved in establishing the traceability for the SEM measurements of nanoparticle size to facilitate the work of future operators. But before the reviewer recommends this work to be published in nanomaterials, some issues need to be tackled.

  1. The full text lacks innovation in scientific paper writing, and the experimental characteristics are more similar to patents.

After a thorough search in scientific literature, we realised that the articles concerning the dimensional metrology of nanoparticles are quite rare. However, the measurement of size is a key stage in nanoparticle research and uncertainties need to be assessed to give a level of confidence in this measurement. Calibration and uncertainty assessment make comparable the measurements and give a universal value to the data. This assessment is based on a qualification of the instrument and numerous experimental data are required. To this end, it is essential to have a perfect understanding not only of how the instrument works, but also of the physical principles involved behind. Our aim in this article is to describe all this philosophy to the reader. According to us, this should not be the subject of a patent but rather of a scientific article. Furthermore, this is the first time that a curve of measured size / EHT is observed with a similar behaviour for all studied nanoparticles and we demonstrate how this curve can be used for calibrating the instrument. The method proposed here is essentially designed for nanomaterial researchers interested in carrying out size measurements.

  1. As shown in Figure 1, to highlight the differences in NP behaviour regarding their size and chemical composition during electron/matter interactions induced by the SEM e-beam, four samples of metal, oxide, and polymer nanoparticles were studied. So, why did you choose these four samples for study?

We have selected four samples with very different atomic number to have a better understanding of the interaction e-beam/nanoparticle. Moreover, regarding a similar chemical composition, we have also tried several sizes to highlight the impact of the dimension on the reliability of the measurements. This has enabled us to demonstrate the universal behaviour of the variation curve of the measured size as a function of EHT, whatever the size and chemical composition of the particle.

The first sentence of section 2.1 (page 4) was changed. “In order to have a better understanding of the electron / matter interactions induced by the SEM e-beam, four samples of different sizes and atomic numbers (metal, oxide, polymer) were studied. The samples (Au, SiO2, PSL) were chosen because they are widely used in laboratory.”

  1. As shown in Figure 6, when the acceleration voltage of gold nanomaterials is greater than 6 kv, the modal diameter tends to become a constant value. What are the specific reasons accounting for this result?

Your question is extremely interesting. Unfortunately, we have been unable to obtain any further information about the presence of this plateau. Two issues remain, (i) is this plateau linked to metallic feature of gold material? (ii) Do the values of measured diameter increase with higher EHT? We have reason to believe that measured diameters increase above 10 kV because we have some experimental data to show this. However, the model used in this study did not allow us to investigate the gold nanoparticle behaviour above 10 kV. As a result, we have not reliable results up to now. In the near future, our objective is to improve the model in order to study the SEM signal over all EHT range.

  1. Where are relations (16) (17)、(18) and (19)?

Page 12. The numbering error has been corrected.

  1. For materials with poor radiation resistance, how to use this method to accurately measure distances?

Thank you for your relevant comment. Our study is only limited to metal, oxide and polymer. However, we do not know the behaviour of materials that are less resistant to radiation, such as an organic substance, for instance. In this paper, we demonstrated that regarding studied materials the “true value” of the size can be reached at relatively low accelerating voltage (EHT), i.e. smaller than 3 kV. We can guarantee that this range of energy of primary electrons is not sufficient to cause damage the nanoparticles. Nevertheless, we have to keep in mind that the properties of substrate are also determinant. Nanoparticle deposited on insulating substrate can be deformed by the e-beam.

  1. How to avoid the influence of current in scanning electron microscopes?

Your question is relevant. Theoretically, the current have no impact on the measurement. Unfortunately, we cannot check this because this parameter cannot be adjusted with our current instrument. The value indicated on the control interface is for guidance only. Our microscope is not equipped with a Faraday cage for accurately determining the current. However, in the near future, the nanoparticle size measurement will be carried out with a new microscope, installed very recently at laboratory, and this parameter will be able to be investigated more precisely. The potential impact of current will be compared with our model as well. According to us, the current should just affect the size of the probe (e-beam diameter). A high current could potentially accentuate charging effects, while a low current would degrade the signal-to-noise ratio. All this will be verified in the near future.

Comments on the Quality of English Language. It can be improved.

The text has been proofread by a native Englishwoman and corrections have been made.

Reviewer 2 Report

Comments and Suggestions for Authors

The authors in the manuscript propose a general evaluation method of the uncertainties for measuring nanoparticle size by SEM, focusing mainly on the investigation of the impact of the accelerating voltage (EHT) on the reliability of the size measurements (section 4.3), using samples of silica, polystyrene latex and gold nanoparticles. 

The study is presented in a straightforward manner and the amount of work that is involved in this manuscript is sufficient. The authors have made a fair number of references to relevant literature. The manuscript is suitable for publication if the authors address the following minor issues:

1.     Page 11 line 28: the relation numbers should be corrected

2.  Page 11 line 30: the sequence of the equation numbers is not correct (equation 20 on line 30 follows equation 15 on line 26)

3.    Sections 4.2.3 and 4.2.4 should be revised and presented in more detail to facilitate reading.

4. Page 13 lines 28-30: The sentence is difficult to follow and should be rephrased.

5.  Page 16, Figure 10: The authors should use different symbols in the profile curves  

Author Response

The authors in the manuscript propose a general evaluation method of the uncertainties for measuring nanoparticle size by SEM, focusing mainly on the investigation of the impact of the accelerating voltage (EHT) on the reliability of the size measurements (section 4.3), using samples of silica, polystyrene latex and gold nanoparticles.

The study is presented in a straightforward manner and the amount of work that is involved in this manuscript is sufficient. The authors have made a fair number of references to relevant literature. The manuscript is suitable for publication if the authors address the following minor issues:

The authors are very grateful to the referee for his encouraging comments.

  1. Page 11 line 28: the relation numbers should be corrected
  2. Page 11 line 30: the sequence of the equation numbers is not correct (equation 20 on line 30 follows equation 15 on line 26)

Remarks 1 and 2: Page 12. The numbering error has been corrected.

  1. Sections 4.2.3 and 4.2.4 should be revised and presented in more detail to facilitate reading.

The authors would like to thank you for this comment. These two sections have been clarified and errors corrected. In fact, some inconsistencies linked to variables have been removed from the equations.

  1. Page 13 lines 28-30: The sentence is difficult to follow and should be rephrased.

It is true that the sentence was not clear, thanks a lot for that. It was rephrased by: “It has been noticed that here the signal to noise ratio is not calculated from equation (1), but estimated only as the ratio of nanoparticle signal over the average of the substrate signal.”

  1. Page 16, Figure 10: The authors should use different symbols in the profile curves

The Figure has been modified by using different symbols.

Reviewer 3 Report

Comments and Suggestions for Authors

The manuscript under consideration is devoted to a problem that is common for many researchers that are working with nanomaterials. That is how to precisely define the lateral dimensions of object imaged with a scanning electron microscope. In other words, they describe the SEM calibration procedure. The topic is important, and the results presented in the manuscript are convincing. The authors come up with description of a calibration protocol that is based on previous results in the field of scanning electron microscopy as well as those obtained by authors.

Still, I suggest to resubmit the paper to MDPI /Instruments/ journal rather than publish it the Nanomaterial journal. The manuscript provides a lot of technical information relevant to protocols of the SEM calibration and use as a metrological instrument. There is next to no new information related directly to material science. At the same time physics of the interaction of electron beam with different materials and structures is considered just qualitatively.

I recommend to address the following issues.

Language, clarity of presentation

The certificate 12 gives an indicative value for electron microscopy techniques of 27.8 nm with 1.5 nm associated uncertainty (k=2)[17]

What is k? Definition of this parameter has to be provided.

This detector is located just above the sample at the end of the column and is particularly 4 well suitable for measuring the properties of nanoparticles.

What makes this particular detector better than, e.g. SE2 detector?

The drift of the sample stage was also investigated during 1 and 10 minutes [21]. The time taken to acquire images is smaller than 30 seconds, so the stage stability is sufficient not to add this kind of drift in the final uncertainty.

What is the relationship between the time in ref 21 and image acquisition? What if one studied the drift during 1 second? Would it impose a different restriction?

This phenomenon is due to the acceleration of the e-beam required at the beginning of every line scan

NOT beam itself but its deflection

The curve first decreases in the small 23 voltage range (EHT <2-3 kV) and reaches a minimum.

Very bad language. A quantity or a value can decrease or increase. Not the curve

Influence of the substrate material, nanoparticle material and the accelerating voltage on the image is discussed in the manuscript but very qualitatively.

Questions to be addressed

Maximal magnification used in the fig. 3 is about 20 to 30 thousand times while the data displayed is extrapolated to 100 thousand times which is not justified by the experimentally verified data. On the other hand magnification of 100 thousand times is not sufficient for certain tasks and calibration for magnification u to 300 thousand times should be carried out

SEM magnifications have to be given for all images used in the figures.

Role of the scanning rate is not discussed at all.

Comments on the Quality of English Language

Quality of English is quite satisfactory.

Author Response

The manuscript under consideration is devoted to a problem that is common for many researchers that are working with nanomaterials. That is how to precisely define the lateral dimensions of object imaged with a scanning electron microscope. In other words, they describe the SEM calibration procedure. The topic is important, and the results presented in the manuscript are convincing. The authors come up with description of a calibration protocol that is based on previous results in the field of scanning electron microscopy as well as those obtained by authors.

Still, I suggest to resubmit the paper to MDPI /Instruments/ journal rather than publish it the Nanomaterial journal. The manuscript provides a lot of technical information relevant to protocols of the SEM calibration and use as a metrological instrument. There is next to no new information related directly to material science. At the same time physics of the interaction of electron beam with different materials and structures is considered just qualitatively.

The authors are grateful for your advice. But, according to us, calibrating a microscope and assessing uncertainties are an integral part of the measurement procedure carried out by a specialist in the study of nanomaterials. The results presented in the text are qualitative but they were obtained through a model described in a previous article. According to us, the results concerning the evolution of the size measured with EHT are quantitative but are determining for the development of the nanomaterials. They demonstrate the impact of electron/nanoparticle interaction on the size measurements and the material experts must know this kind of information.

I recommend to address the following issues.

Language, clarity of presentation

The certificate 12 gives an indicative value for electron microscopy techniques of 27.8 nm with 1.5 nm associated uncertainty (k=2)[17]

What is k? Definition of this parameter has to be provided.

Page 4 and others: k is the coverage factor. The expanded intervals correspond to a 95 % coverage probability. This clarification has been added when it first referenced page 4.

This detector is located just above the sample at the end of the column and is particularly 4 well suitable for measuring the properties of nanoparticles.

What makes this particular detector better than, e.g. SE2 detector?

Page 5. Thank you for suggesting this clarification point. The following sentence has been added: “In-Lens detector collects SE generated close to the surface, and in the context of our study, making it particularly sensitive to nano-object deposited on a substrate in contrast to classical SE2 detector placed in lateral position of the chamber.”

The drift of the sample stage was also investigated during 1 and 10 minutes [21]. The time taken to acquire images is smaller than 30 seconds, so the stage stability is sufficient not to add this kind of drift in the final uncertainty.

What is the relationship between the time in ref 21 and image acquisition? What if one studied the drift during 1 second? Would it impose a different restriction?

This phenomenon is due to the acceleration of the e-beam required at the beginning of every line scan

NOT beam itself but its deflection

Page 9. You are perfectly right. It is a misuse of language. The sentence have been replaced by “When scanning a new line, the beam is deflected to change direction leading to a variation in scan speed at the beginning of each line”.

The curve first decreases in the small 23 voltage range (EHT <2-3 kV) and reaches a minimum.

Very bad language. A quantity or a value can decrease or increase. Not the curve

Page 13. The word “curve” has been replaced with “Values”

Influence of the substrate material, nanoparticle material and the accelerating voltage on the image is discussed in the manuscript but very qualitatively.

Questions to be addressed

Maximal magnification used in the fig. 3 is about 20 to 30 thousand times while the data displayed is extrapolated to 100 thousand times which is not justified by the experimentally verified data. On the other hand magnification of 100 thousand times is not sufficient for certain tasks and calibration for magnification u to 300 thousand times should be carried out

Page 7 and 8, Figure 3, table 2. Thank you for your comment. We fully agree with you, this point must be addressed. Data beyond magnification higher than 25kX had been extrapolated and the range in the figure will be reduced to avoid a misinterpretation. Since standard with lower pitch is not available at LNE, and we want to comply with the ISO standard, we cannot go any further for calibration in terms of magnification. We've added a paragraph that mentions the ISO rules for evaluating pitches of grating with Fast Fourier Transform that we are using for automatically measure the pitch on an image to explain the constraints that led us to this decision. Beyond a magnification of 25 000, measurements of certified nanoparticles are required for verification.

SEM magnifications have to be given for all images used in the figures.

Magnifications were added in captions for all images.

Role of the scanning rate is not discussed at all.

Page 7. This uncertainty source had been evaluated in a previous article, given in ref [21]. This source is negligible as others sources like brightness or contrast. A sentence has been added, page 7 ” These sources of uncertainty include repeatability measurements, magnification, beam width, operator, pixel size, image analysis, contrast, brightness, scan speed, drifts, noise reduction parameter, focus and astigmatism adjustments. Some of these are considered negligible and are not covered in this article (scan speed, contrast, brightness, focus and astigmatism adjustment).”

Comments on the Quality of English Language

Quality of English is quite satisfactory.

The text has been proofread by a native Englishwoman and corrections have been made.

Reviewer 4 Report

Comments and Suggestions for Authors

Nicolas Feltin et al. presented a general evaluation method of the uncertainties for measuring nanoparticle size by SEM, with a specific focus on the investigation of the impact of the energy of the primary electrons generated by e-beam or accelerating voltage on the reliability of the size measurements. Moreover, they also showed that the relationship between size and acceleration voltage depends on the chemical nature of the particles as well as their dimensions. Overall, this work contains many detailed standard experimental designs and analyses, this work shall be interesting and insightful for those researchers on nanomaterials. Below are some minor revision suggestions to improve clarity and address specific concerns:

(1)    Page 3 line 8, the authors claim that the environment effect was neglectable due to the vacuum chamber, however, the electron beam state should be dependent on the working pressure, could the authors discuss more for this point?

(2)    I think the SEM was imaged at a consistent zone for comparison. However, I also noticed that the SEM images were not consistent like Figure 8b, we can see a bright spot at the left bottom corner in 3 kV, as well as a bright spot at the upper left corner in 12 kV image, is this sample imaged at the same sample, please also clarify this point.

(3)    Abbreviation issue:

(i)                  Why accelerating voltage was denoted as EHT (page 1 line 25), is this a generalized abbreviation? Please also clarify the issue on page 12 line 18.

(ii)                 Duplicated interpretation for abbreviation like EHT on page 1 line 25, page 12 line 18, page 17 line 30, page 18 line 15; also like TSL on page 12 line 7, page 12 line 15, page 19 line 24. Generally, a one-time interpretation of the abbreviation would be enough when it was first referred.

(iii)               Page 3 line 22, what does EM represent? Page 5 line 17, WD denotes working distance?

(iv)               Page 4 Figure 1, what does the number 40/80 nm refer? Counting the particle size within this range? Why there are many small-sized particles below 90 nm in the PSL-90 nm sample?

(v)                 The k value shall also be denoted a better understanding for those researchers beyond electron microscopy line page 4 line 10, page 5 line 17, page 7 line 4, and others.  Additionally, some equations shall be further interpreted line page 10 line 44, what does b refer?

(4)    I understand that the size measurement above 50 nm would be reliable via SEM, however, many nanoparticles reveal new properties below 20 nm or even 10 nm, like quantum dots (doi.org/10.1002/EXP.20220169) or nanoflakes (doi.org/10.1038/s41563-020-00831-1) for energy storage devices, photonics, catalysis, and others. Could the authors discuss more at the end of this paper, any special attention was required for the measuring and imaging of low-dimensional nanomaterials with small sizes via SEM. This would be inspiring and important for researchers with broad interests.

Author Response

Nicolas Feltin et al. presented a general evaluation method of the uncertainties for measuring nanoparticle size by SEM, with a specific focus on the investigation of the impact of the energy of the primary electrons generated by e-beam or accelerating voltage on the reliability of the size measurements. Moreover, they also showed that the relationship between size and acceleration voltage depends on the chemical nature of the particles as well as their dimensions. Overall, this work contains many detailed standard experimental designs and analyses, this work shall be interesting and insightful for those researchers on nanomaterials. Below are some minor revision suggestions to improve clarity and address specific concerns:

(1)  Page 3 line 8, the authors claim that the environment effect was neglectable due to the vacuum chamber, however, the electron beam state should be dependent on the working pressure, could the authors discuss more for this point?

Page 3. The authors thank the referee for this comment. This allows us to clarify this point. In the case of an environmental SEM (Low vacuum system), this question should be discussed. At LNE, the Zeiss Ultra Plus microscope works only in vacuum, and the level of vacuum is assumed to be constant (5.10-6 mbar). The sample is introduced trough a load-lock chamber. Moreover, to avoid any issue of drift and contamination, the measurements are always realized with at least one night pumping the chamber (page 5). This clarification has been added in the text.

(2) I think the SEM was imaged at a consistent zone for comparison. However, I also noticed that the SEM images were not consistent like Figure 8b, we can see a bright spot at the left bottom corner in 3 kV, as well as a bright spot at the upper left corner in 12 kV image, is this sample imaged at the same sample, please also clarify this point.

For technical reasons, images could not be performed on the same population of nanoparticles at different EHT. In Figure 8 a), b) and Figure 9 a), b), the series of images correspond to different nanoparticles but with exactly the same size. On the other hand, the Figures 10 a) and 11 b) represent two single particles, the one deposited on Si wafer (Fig. 10) and the other on carbon membrane (Fig. 11), respectively, submitted at different EHT.

(3) Abbreviation issue:

(i) Why accelerating voltage was denoted as EHT (page 1 line 25), is this a generalized abbreviation? Please also clarify the issue on page 12 line 18.

Yes, accelerating voltage is very often denoted EHT (Electron High Tension) or named landing energy. However, on page 14 line 18, we specified that EHT and landing energy might be slightly different if the material charges and becomes electro-negative or if a deceleration is applied on the e-beam. It is not the case in our study. We decided to delete EHT and keep “accelerating voltage” in the abstract to avoid any confusion.

(ii) Duplicated interpretation for abbreviation like EHT on page 1 line 25, page 12 line 18, page 17 line 30, page 18 line 15; also like TSL on page 12 line 7, page 12 line 15, page 19 line 24. Generally, a one-time interpretation of the abbreviation would be enough when it was first referred.

Duplicated interpretation of the abbreviation were deleted as mentioned.

(iii) Page 3 line 22, what does EM represent? Page 5 line 17, WD denotes working distance?

“EM” means Electron Microscopy. Acronyms EM and WD (Working Distance) have been defined in the introduction and section 2.2, respectively.

(iv) Page 4 Figure 1, what does the number 40/80 nm refer? Counting the particle size within this range? Why there are many small-sized particles below 90 nm in the PSL-90 nm sample?

Thanks a lot for this comment. We have clarified this point. The sample FD101b is composed of a bimodal population of nanoparticles with a mode at roughly 40 nm and 80 nm. However, only the value of 80 nm nominal mode is certified for Electron Microscopy. The sample is now correctly described in section 2.1. The PSL sample also consists of bimodal population but only 90 nm peak is considered in the text. A brief description of this sample was added in section 2.1.

(v) The k value shall also be denoted a better understanding for those researchers beyond electron microscopy line page 4 line 10, page 5 line 17, page 7 line 4, and others. Additionally, some equations shall be further interpreted line page 10 line 44, what does b refer?

In metrology, k denotes the coverage factor. For instance, k=2 means the expanded interval corresponding to a 95 % coverage probability. This information was added within the text. b is the potential bias corresponding to an instrument drift observed between two calibrations. This value is now defined in section 4.2.

(4) I understand that the size measurement above 50 nm would be reliable via SEM, however, many nanoparticles reveal new properties below 20 nm or even 10 nm, like quantum dots (doi.org/10.1002/EXP.20220169) or nanoflakes (doi.org/10.1038/s41563-020-00831-1) for energy storage devices, photonics, catalysis, and others. Could the authors discuss more at the end of this paper, any special attention was required for the measuring and imaging of low-dimensional nanomaterials with small sizes via SEM. This would be inspiring and important for researchers with broad interests.

Microscopists usually say that it is very difficult to measure nano-objects with sizes smaller than 10 nm with SEM. Below 10 nm, the pixel size is too large in comparison with the NP size. Consequently, this paper deals with the dimension measurements of nanoparticles ranging from 10 nm to 100 nm. This point was specified in the conclusion.

Reviewer 5 Report

Comments and Suggestions for Authors Nanometrology plays a crucial role in the manufacture of nanomaterials and highly accurate and reliable nanodevices. Measure the size of nanoparticles is therefore of paramount importance. However, size determination by electron microscopy-based techniques often remains a real challenge in many cases. Establishing traceability of measurements, which is essential to ensure the reliability of data and the comparability of measurement results, is also often overlooked. In this work, authors propose a method for ensuring the traceability of nanoparticle size measurements by SEM. And the effect of primary electron (PE) energy generated by an electron beam or accelerating voltage (EHT) on the reliability of dimensional measurements is explored in depth. In summary, this work is comprehensive and interesting. However, there are some comments must be addressed before this paper publication. 1. The first appearance of abbreviation should be defined, e.g., ‘e-beam’ on page 1, line 25. 2. Abbreviations only need to be defined the first time they appeared, not repeatedly. Please check and correct throughout the article. 3. Please check font formatting for proper bolding, e.g., ‘(1)’ on page 5, line 35; ‘Figure 3’ on page 8, line 24. 4. Please provide the relations (16), (17), (18) and (19) mentioned in the manuscript. 5. Figures are an important part of helping readers understand the content of an article, and their readability and aesthetic layout should be ensured. (1) Lack of description of Figure 7, please added a brief descriptions for better understanding. (2) The font size of the horizontal and vertical coordinates should be uniform across the subplots. Comments on the Quality of English Language

Please ask a native english speaker polish the english

Author Response

Nanometrology plays a crucial role in the manufacture of nanomaterials and highly accurate and reliable nanodevices. Measure the size of nanoparticles is therefore of paramount importance. However, size determination by electron microscopy-based techniques often remains a real challenge in many cases. Establishing traceability of measurements, which is essential to ensure the reliability of data and the comparability of measurement results, is also often overlooked. In this work, authors propose a method for ensuring the traceability of nanoparticle size measurements by SEM. And the effect of primary electron (PE) energy generated by an electron beam or accelerating voltage (EHT) on the reliability of dimensional measurements is explored in depth. In summary, this work is comprehensive and interesting. However, there are some comments must be addressed before this paper publication.

The authors are very grateful to the referee for his encouraging comments.

  1. The first appearance of abbreviation should be defined, e.g., ‘e-beam’ on page 1, line 25.

Abbreviation e-beam is defined in the introduction and “electron beam” used in abstract.

  1. Abbreviations only need to be defined the first time they appeared, not repeatedly. Please check and correct throughout the article.

All abbreviations are now correctly defined at the beginning of the article and are used hereafter.

  1. Please check font formatting for proper bolding, e.g., ‘(1)’ on page 5, line 35; ‘Figure 3’ on page 8, line 24.

Le formatage de la police pour une mise en gras correcte a été effectué.

  1. Please provide the relations (16), (17), (18) and (19) mentioned in the manuscript.

The numbering error has been corrected.

  1. Figures are an important part of helping readers understand the content of an article, and their readability and aesthetic layout should be ensured

(1) Lack of description of Figure 7, please added a brief descriptions for better understanding.

The caption of Figure 7 was modified for further details “Histograms of size distribution of the four studied samples (FD-101b, FD-304, PSL and BBI nano-gold) at different acceleration voltage (EHT) ranged from 1 kV to 10 kV. The value of the modal diameter is given for each histogram to highlight the impact of the EHT parameter on the measured size.”

(2) The font size of the horizontal and vertical coordinates should be uniform across the subplots.

The figure has been modified in this way

Comments on the Quality of English Language

Please ask a native english speaker polish the English

The text has been proofread by a native Englishwoman and corrections have been made.

Round 2

Reviewer 1 Report

Comments and Suggestions for Authors

All the concerns we proposed had been properly addressed. 

Author Response

All the concerns we proposed had been properly addressed.

LNE’s Nanometrology team warmly thanks the referee for the fruitful comments.

Reviewer 3 Report

Comments and Suggestions for Authors

The manuscript under discussion is improved after my specific comments were accounted for. But there is a fundamental problem with this paper. It comes up with a protocol that is supposed to allow for precise definition of the lateral dimensions of objects imaged with a scanning electron microscope. Although supported by previous results in the field of scanning electron microscopy as well as those obtained by authors, the protocol is far from being universal. It can be applied for certain range of parameters, such as nanoparticle material, substrate material, etc. Still, no restrictions to its applicability are discussed. Such a paper should be addressed to the members of the community of those who have expertise in scanning electron microscopy.

On my view, a paper addressed to the Nanomaterial readers should not discuss specific protocols but rather their applicability to this or that materials and situations.

Not being an expert in electron microscopy, I can come up with at least one example for which the technique described in the paper under consideration does not work. That is carbon nanotubes on a Si/SiO2 wafer. Or on PMMA. Or any other dielectric substrate.

Brintlinger, T., et al. "Rapid imaging of nanotubes on insulating substrates." Applied Physics Letters 81.13 (2002): 2454-2456.

Homma, Yoshikazu, et al. "Mechanism of bright selective imaging of single-walled carbon nanotubes on insulators by scanning electron microscopy." Applied Physics Letters 84.10 (2004): 1750-1752.

I did not mention this in my first review since it took me some time to “dig out” this example that I ran into some long time ago working with CNTs. It does not mean that other restrictions will apply to applicability of the procedure suggested by authors. This is why I suggest transferring this manuscript to a journal addressing more narrow community of researchers, i.e. readership of the MDPI /Instruments/ journal.

Again, the physics of electron beam interaction with nanoparticles is discussed in a very primitive fashion. For example, the dependence of the “nanoparticle size” on the EHT value should be analyzed based on the EHT dependence of the secondary electron yield which is different for different materials. See, for example Rau, E. I., et al. "Second crossover energy of insulating materials using stationary electron beam under normal incidence." Nuclear Instruments and Methods in Physics Research Section B: Beam Interactions with Materials and Atoms 266.5 (2008): 719-729.

The explanation on why in-Lense detector should be used rather than the SE2 one is very primitive, stating that “In-Lens detector collects SE generated close to the surface, and in the context of our study, making it particularly sensitive to nano-object deposited on a substrate in contrast to classical SE2 detector placed in lateral position of the chamber.”

What is “close”? It should be “small enough to …” or “smaller than …” Just “close” is meaningless in physics. And where are generated the secondary electrons detected by the SE2? What is the ratio between the distances?

All in all, in my opinion, neither the scope nor the scientific impact of the paper fit the Nanomaterials journal.

Author Response

The manuscript under discussion is improved after my specific comments were accounted for. But there is a fundamental problem with this paper. It comes up with a protocol that is supposed to allow for precise definition of the lateral dimensions of objects imaged with a scanning electron microscope. Although supported by previous results in the field of scanning electron microscopy as well as those obtained by authors, the protocol is far from being universal. It can be applied for certain range of parameters, such as nanoparticle material, substrate material, etc. Still, no restrictions to its applicability are discussed. Such a paper should be addressed to the members of the community of those who have expertise in scanning electron microscopy.

We can understand your concerns, but we have never described the protocol as universal. The restrictions are present in the title. The paper deals with the measurements of size only applied for nanoparticles. According to us, this study is not addressed only to the experts in Microscopy, and it is important that anyone involved with nanomaterials who has to carry out size measurements should be familiar with this protocol. We have already shared this approach with other colleagues and it has been very well received.

On my view, a paper addressed to the Nanomaterial readers should not discuss specific protocols but rather their applicability to this or that materials and situations.

As far as we know, this protocol can be applied to any nanoparticle. Furthermore, any researcher involved in the study of nanoparticles should know the basics of metrology and be aware of the importance of traceability. After a comprehensive survey in the literature, we have not found clear articles about the traceability impact on the nanoparticle size measurement accuracy.

Not being an expert in electron microscopy, I can come up with at least one example for which the technique described in the paper under consideration does not work. That is carbon nanotubes on a Si/SiO2 wafer. Or on PMMA. Or any other dielectric substrate.

Brintlinger, T., et al. "Rapid imaging of nanotubes on insulating substrates." Applied Physics Letters 81.13 (2002): 2454-2456.

Homma, Yoshikazu, et al. "Mechanism of bright selective imaging of single-walled carbon nanotubes on insulators by scanning electron microscopy." Applied Physics Letters 84.10 (2004): 1750-1752.

I did not mention this in my first review since it took me some time to “dig out” this example that I ran into some long time ago working with CNTs. It does not mean that other restrictions will apply to applicability of the procedure suggested by authors. This is why I suggest transferring this manuscript to a journal addressing more narrow community of researchers, i.e. readership of the MDPI /Instruments/ journal.

I fully agree with you but according to the standardization, nanotubes are not considered to be nanoparticles (“3 external dimensions in the nanoscale”, ISO 80004-1:2023).

Again, the physics of electron beam interaction with nanoparticles is discussed in a very primitive fashion. For example, the dependence of the “nanoparticle size” on the EHT value should be analyzed based on the EHT dependence of the secondary electron yield which is different for different materials. See, for example Rau, E. I., et al. "Second crossover energy of insulating materials using stationary electron beam under normal incidence." Nuclear Instruments and Methods in Physics Research Section B: Beam Interactions with Materials and Atoms 266.5 (2008): 719-729.

You are right, in this paper, we use the results of the model already described and applied on different nanoparticles in the following reference :

Crouzier, L.; Delvallée, A.; Devoille, L.; Artous, S.; Saint-Antonin, F.; Feltin, N. Influence of Electron Landing Energy on the Measurement of the Dimensional Properties of Nanoparticle Populations Imaged by SEM. Ultramicroscopy 2021, 226, 113300, doi:10.1016/j.ultramic.2021.113300.

But, we can also find some information in:

Crouzier, L.; Delvallée, A.; Ducourtieux S., Devoille, L.; Tromas C.; Feltin, N. A new method for measuring nanoparticle diameter from a set of SEM images using a remarkable point. Ultramicroscopy, Volume 207, 2019, 112847.

The explanation on why in-Lense detector should be used rather than the SE2 one is very primitive, stating that “In-Lens detector collects SE generated close to the surface, and in the context of our study, making it particularly sensitive to nano-object deposited on a substrate in contrast to classical SE2 detector placed in lateral position of the chamber.”

What is “close”? It should be “small enough to …” or “smaller than …” Just “close” is meaningless in physics. And where are generated the secondary electrons detected by the SE2? What is the ratio between the distances?

In-Lens detector is well known to the electron microscopy experts. But for non-specialists we have indicated in the previous sentence that “this detector is located just above the sample at the end of the column”. Then, this detector is located just above and is closer to the sample than the SE2 detector, which is positioned on the lateral side of the chamber. It is more sensitive to secondary electrons coming from the surface, called SE1. The nanometric size of nanoparticles explains why they can be considered part of the surface. Therefore, In-Lens gives more information regarding the size of nanoparticles. We do not know the ratio between In-Lens and SE2 distances, but the ability of In-Lens to collect more specifically secondary electrons generated by a nano-object (SE1) in the whole group of secondary electrons compared to the SE2 detector is well known to microscopists.

All in all, in my opinion, neither the scope nor the scientific impact of the paper fit the Nanomaterials journal.

Once again, in our opinion, the basics of a dimensional metrology approach for measuring nanoparticles should be familiar to any researcher involved in this field. To us, this paper is in keeping with the generalist nature of Nanomaterials journal.
